# Circuit Simulation Considering Electrical Coupling in Monolithic 3D Logics with Junctionless FETs

**DOI:** 10.3390/mi11100887

**Published:** 2020-09-24

**Authors:** Tae Jun Ahn, Yun Seop Yu

**Affiliations:** 1Department of Electrical, Electronic and Control Engineering and IITC, Hankyong National University, 327 Jungang-ro, Anseong-si, Gyenggi-do 17579, Korea; jigo1235@hknu.ac.kr; 2Group for Smart energy Nano Convergence, Korea Institute of Industrial Technology, 6, Cheomdangwagi-ro 208 beon-gil, Buk-gu, Gwangju 61012, Korea

**Keywords:** junctionless FET, JLFET, electrical coupling, circuit simulation, parameter extraction, monolithic 3D integrated circuit (IC)

## Abstract

The junctionless field-effect transistor (JLFET) compact model using the model parameters extracted from the LETI-UTSOI (version 2.1) model was proposed to perform circuit simulation considering the electrical coupling between the stacked JLFETs of a monolithic 3D integrated circuit (M3DIC) composed of JLFETs (M3DIC-JLFET). We validated the model by extracting the model parameters and comparing the simulation results of the technology computer-aided design and the Synopsys HSPICE circuit simulator. The performance of the M3DIC-JLFET was compared with that of the M3DIC composed of MOSFETs (M3DIC-MOSFET). The performance of a fan-out-3 ring oscillator with M3DIC-JLFET varied by less than 3% compared to that with M3DIC-MOSFET. The performances of ring oscillators of M3DIC-JLFET and M3DIC-MOSFET were almost the same. We simulated the performances of M3DICs such as an inverter, a NAND, a NOR, a 2 × 1 multiplexer, and a D flip-flop. The overall performance of the M3DIC-MOSFET was slightly better than that of the M3DIC-JLFET.

## 1. Introduction

Monolithic 3-dimensional integration (M3DI) refers to a 3D integration scheme of sequentially manufacturing and stacking devices [1,2,3]. M3DI has been studied extensively as an alternative to improve semiconductor performance in a region where the scale-down limit of a semiconductor device is approaching. In memories (e.g., NAND flash and dynamic random-access memory) and sensors (e.g., 3D heterogeneous integration), the sequential stacking M3DI method has already been applied instead of the through-silicon via method [4,5,6,7]. In addition, studies have reported that the performance of electrical coupling improves when the inter-layer dielectric (ILD) thickness of the M3D complementary metal-oxide-semiconductor logic is less than 50 nm [8]. M3DI in terms of logic has the potential to enhance chip performance, interconnect delay, device density, and frequency bandwidth without requiring the further lateral scaling of the device [9]. Owing to the process for device stacking sequentially on a single wafer, the previous and next tiers have significant limitations in the process thermal budget for device quality [10]. Compared to the conventional standard process, low-temperature processes using approximately 650 °C have been developed, improving the performance of M3DI [11,12,13]. Currently, most M3DI devices have been researched based on metal-oxide-semiconductor field-effect transistors (MOSFETs) that use Si, Ge, and III-V materials [14,15,16,17]. For the majority of MOSFETs, a thermal budget is required for dopant activation after the implantation process; however, there are physical limitations for using these as low-power devices. For junctionless field-effect transistors (JLFETs), it is possible to use the MOSFET process as it has a junctionless structure. This means that dopant activation is not required, unlike in MOSFETs. JLFETs are advantageous for scale-down, surface mobility degradation, and short-channel effects [18]. A new circuit simulation model has been proposed in which the M3DIC composed of MOSFETs (M3DIC-MOSFETs) reflect direct current (DC)/alternating current (AC) and transient inter-layer electrical coupling [19]. However, owing to the absence of a JLFET compact model that considers electrical coupling between the tiers for the circuit simulation of M3DI structures, an accurate circuit simulation for M3DICs is not possible [20,21,22,23].

In this study, to extract the parameters of the model for the circuit simulation of the M3DIC-JLFET, a structure of monolithic 3D inverter (M3DINV) with electrical coupling (thickness of ILD, *T_ILD_* = 10 nm) was constructed and simulated using technology computer-aided design (TCAD). To perform circuit simulation considering the electrical coupling of M3DIC-JLFET, we propose the LETI-UTSOI (version 2.1) model [24,25,26] of the fully-depleted silicon-on-insulator (FD-SOI) MOSFET structure as an alternative to the JLFET compact model and extract the model parameters. The extracted model parameters were verified and compared to the TCAD mixed-mode simulation results (Section 3). Based on the model parameters extracted in Section 3, various logics were simulated, and the performance was compared with that of M3DIC-MOSFETs (Section 4) [27]. Section 5 concludes this study.

## 2. Structures

Figure 1 shows the schematics of an M3DINV composed of JLFET (M3DINV-JLFET). As shown in Figure 1a, an M3DINV-JLFET consists of n-type and p-type JLFET transistors in the top and bottom tiers, respectively. The doping of the JLFET’s source/drain, lightly-doped drain (LDD), and the channel are 10^20^, 10^20^, and 10^19^ cm^−3^, respectively. The JLFET was simulated at the gate length (*L_g_*), gate oxide film (*T_ox_*), silicon thickness (*T**_si_*), and ILD thickness (*T_ILD_*) at 30, 1, 6 m, and 10 nm, respectively. The gate oxide, ILD, and box were composed of SiO_2_.

In reference data simulation, a device simulator, ATLAS [28], by SILVACO was used. Table 1 shows the models, methods, and work functions used in the TCAD simulation. The models used for the device simulation were CVT, SRH, BGN, AUGER, and FERMI. The methods used for device simulation were NEWTON and GUMMEL. The gate work functions of the n-type and p-type JLFETs were 5.06 and 4.41 eV, respectively.

## 3. Parameter Extraction

Figure 2 shows the Simulation Program with Integrated Circuit Emphasis (SPICE) model parameter extraction process used in this study. Through the process flow, model parameters were extracted by comparing them with the reference data. First, parameter initialization was performed, and the threshold voltage (*V_t_*) roll-off and subthreshold swing (*SS*) degradation parameters were extracted. Next, the mobility and series resistance parameters, velocity saturation, drain-induced barrier lowering (DIBL), and channel length modulation (CLM) parameters were extracted. The process was repeated until the parameters were completely extracted. When the DC parameter extraction was complete, the output conductance parameters were extracted. Finally, the back gate effect parameters were extracted.

Figure 3 shows the current-voltage characteristics with the TCAD and HSPICE simulation results of the bottom p-type JLFET. Based on the driving voltage of the inverter, the gate voltage and the drain voltage were verified up to 1 V. Figure 3a shows the drain current-gate voltage (*I*_pds_−*V*_pgs_) characteristics at *V*_p__d__s_ (−0.2, −0.6, and −1 V) and *V*_sub_ = 0 V. Figure 3b shows the *I*_pds_−*V*_p__d__s_ characteristics at *V*_pgs_ (−0.2, −0.6, and −1 V) and *V*_sub_ = 0 V. The HSPICE results match the TCAD results within 10% error.

Figure 4 shows the current-voltage characteristics with the TCAD and HSPICE simulation results of the top n-type JLFET. Figure 4a shows the drain current–gate voltage (*I*_n__ds_−*V*_n__gs_) characteristics at *V*_nd__s_ (0.2, 0.6, and 1 V) and *V*_pg__s_ = 0 V. Figure 4b shows the *I*_n__ds_−*V*_nd__s_ characteristics at *V*_n__gs_ (0.2, 0.6, and 1 V) and *V*_pg__s_ = 0 V. The HSPICE results match the TCAD results within 10% error.

Figure 5a shows the *I*_nds_−*V*_ngs_ characteristics at *V*_pgs_ (0, 0.5, and 1 V) and *V*_nds_ = 1 V. Figure 5b shows the *I*_nds_−*V*_nds_ characteristics at *V*_ngs_ (0.2, 0.6, and 1 V) and *V*_pgs_ = *V*_ngs_. The HSPICE results match the TCAD results within 10% error. This shows that the top n-type JLFET reflects the dependence on the back-gate (gate of the p-type JLFET) bias well. When a voltage is applied to the gate of the bottom p-type JLFET which can operate as a back-gate in M3DINV with a very thin ILD of *T_ILD_* = 10 nm, it affects the current of the top n-type JLFET by the threshold voltage shifts.

Figure 6 shows the transconductance-voltage characteristics of the p-type and n-type JLFET. Figure 6a shows the transconductance-gate voltage (*g_m_*−*V*_pg__s_) characteristics of the bottom p-type JLFET at *V*_pd__s_ (−0.2, −0.6, and −1 V). Figure 6b shows the transconductance-gate voltage (*g_m_*−*V*_ng__s_) characteristics of the top n-type JLFET at *V*_pd__s_ (=0.2, 0.6, and 1 V). We observed a minor mismatch at high gate source voltage values. However, the HSPICE simulation results matched the TCAD results overall within 10% error.

Figure 7 compares the capacitance-voltage characteristics of the p-type and n-type JLFET. Figure 7a shows the gate capacitance-gate voltage (*C*_pgpg_−*V*_pg__s_) characteristics of the bottom p-type JLFET at *V*_pd__s_ (−0.2, −0.6, and −1 V). Figure 6b shows the gate capacitance-gate voltage (*C*_ngng_−*V*_ng__s_) characteristics of the top n-type JLFET at *V*_pd__s_ (0.2, 0.6, and 1 V). The HSPICE simulation results match the TCAD results.

Following the model parameter extraction process flow, we extracted the parameters of the bottom p-type and top n-type JLFET using the LETI-UTSOI model, as shown in Table 2.

Figure 8a shows an equivalent circuit of the M3DINV composed of the top n-type and bottom p-type JLFETs in series. The LETI-UTSOI model was applied to both the JLFETs. The input voltage (*V*_IN_ = *V*_pg_ = *V*_ng_) of M3DINV was applied to the gates of the n-type and p-type JLFETs. The driving voltage (*V*_DD_) was applied to the source of the p-type JLFET, and the source of the n-type JLFET was connected to the ground. The output voltage (*V*_OUT_ = *V*_pd_ = *V*_nd_) was the drain voltage of the n-type and p-type JLFET. Figure 8b compares the voltage transfer characteristics (VTC) on the M3DINV-JLFET, as shown in Figure 8a. The HSPICE simulation results match the TCAD results overall within 10% error.

Figure 9 shows the transient response of the M3DINV-JLFET. Black squares (and solid lines) and red circles (and dot lines) denote the input voltages *V*_IN_ and the output voltages *V*_OUT_ of the M3DINV, respectively. Load capacitance *CL* = 1 fF was used. The HSPICE simulation results match the TCAD results overall within 10% error.

## 4. Circuit Simulation and Discussion

Table 3 shows the power consumption and performance of a fan-out-3 (FO3) ring oscillator built using M3DINV-JLFETs. The M3DINV-JLFETs were compared with the M3DINV-MOSFETs. The power consumption, frequency, and delay per stage of the ring oscillators with 3, 19, and 101 stages of the M3DINV-JLFETs varied less than 3% from those of the M3DINV-MOSFETs. However, the performances of the M3DINV-MOSFET and M3DINV-JLFET were approximately the same.

Table 4 summarizes the performance comparison of M3DIC-JLFETs and M3DIC-MOSFETs. M3DICs such as the INV, NAND, NOR, 2 × 1 multiplexer (MUX) [29], D flip-flop (D-FF) [30], and 6T SRAM [31] were simulated. Their performances were compared in terms of their average static power, average dynamic power, and average delay. The static power of M3DINV-JLFETs increased approximately 600% more than the power of the M3DIC-MOSFETs. Electrical coupling by the gate of the bottom transistor increases the leakage current of the top transistor, resulting in an increase in static power. M3DIC-JLFETs have more leakage current changes due to electrical coupling than M3DIC-MOSFETs. The dynamic power of M3DINV-JLFETs increased approximately 34.5% more than the power of M3DIC-MOSFETs. The average propagation delay of the M3DINV-JLFETs increased approximately 17.5% compared to that of M3DIC-MOSFETs. Because the load cap of M3DIC-JLFETs is larger than that of the M3DIC-MOSFETs due to the electric coupling, the dynamic power and delay of the M3DIC-JLFETs are larger than those of the M3DIC-MOSFETs.

## 5. Conclusions

In this study, we propose to use the LETI-UTSOI (version 2.1) model as an alternative to the JLFET compact model to perform circuit simulation considering the electrical coupling of M3DIC- JLFET. Comparing the simulation results of TCAD and HSPICE, the parameters of the proposed model were extracted and the DC, AC, and transient response characteristics were verified. Although the LETI-UTSOI model of the FD-SOI MOSFET structure is used as an alternative to the JLFET compact model, it was confirmed that circuit simulation considering electrical coupling between vertically stacked JLFETs is possible. Because of the various circuit simulations, the overall performance of the M3DIC-MOSFETs was slightly higher than that of the M3DIC-JLFETs. However, considering the ease of processing, miniaturization, and advantages of M3DI, the applicability of M3DIC-JLFET is higher.

## Figures and Tables

**Figure 1 micromachines-11-00887-f001:**
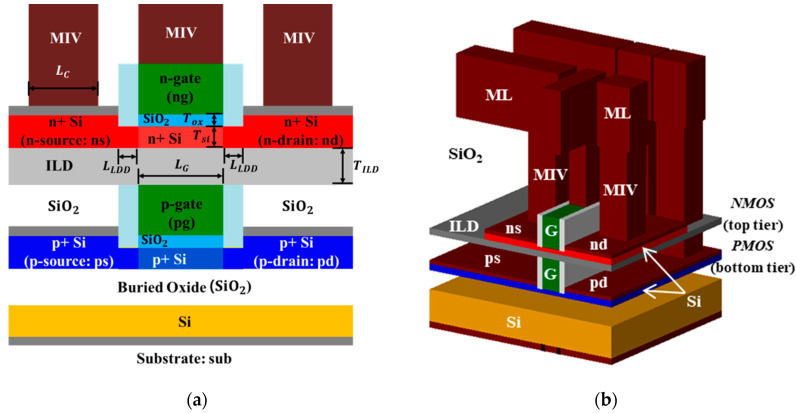
Schematics of two types of M3DINVs composed of JLFET structures: (**a**) 2D cross section; (**b**) 3D structure of M3DINV.

**Figure 2 micromachines-11-00887-f002:**
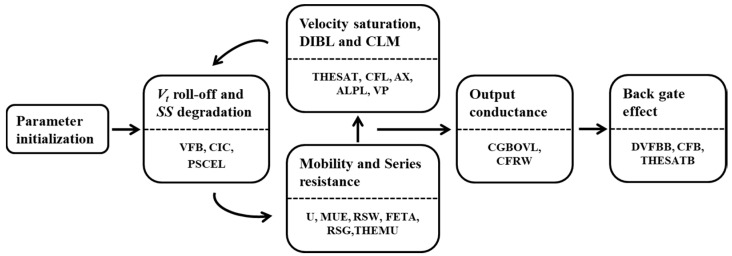
SPICE model’s parameter extraction process flow.

**Figure 3 micromachines-11-00887-f003:**
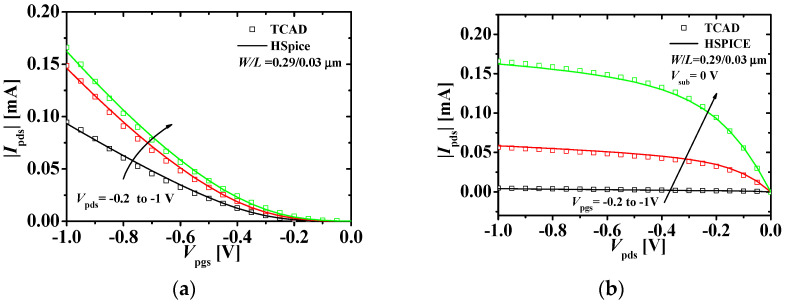
Current-voltage characteristics of the bottom p-type JLFET: (a) |Ipds|−Vpgs characteristics at different values of Vpdss; (b) |Ipds|−Vpds characteristics at different values of Vpgss (squares and lines denote the TCAD and HSPICE simulation results, respectively; W/L = 0.29/0.03 μm)

**Figure 4 micromachines-11-00887-f004:**
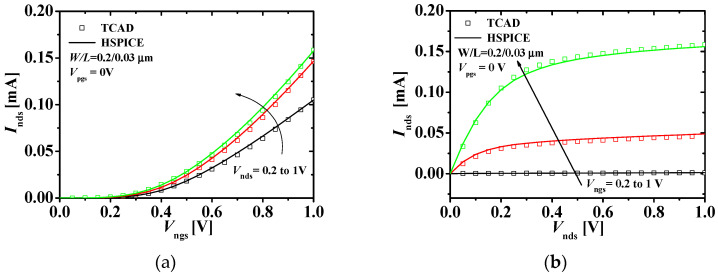
Current-voltage characteristics of the top n-type JLFET: (**a**) *I*_nds_−*V*_ngs_ characteristics at different values of *V*_nds_s; (**b**) *I*_nds_−*V*_nds_ characteristics at different values of *V*_ngs_s at *V*_pgs_ = 0 V (squares and lines denote the TCAD and HSPICE simulation results, respectively; W/L = 0.2/0.03 μm).

**Figure 5 micromachines-11-00887-f005:**
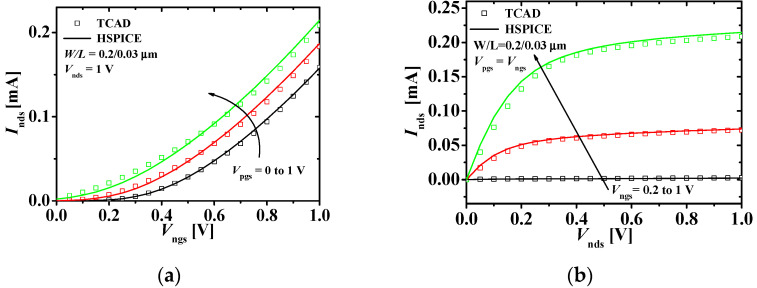
Current-voltage characteristics of the top n-type JLFET: (**a**) *I*_nds_−*V*_ngs_ characteristics at different values of *V*_pgs_s at *V*_nds_ = 0 V; (**b**) *I*_nds_−*V*_nds_ characteristics at different values of *V*_ngs_ s at *V*_pgs_ = *V*_ngs_ (squares and lines denote the TCAD and HSPICE simulation results, respectively; W/L = 0.2/0.03 μm).

**Figure 6 micromachines-11-00887-f006:**
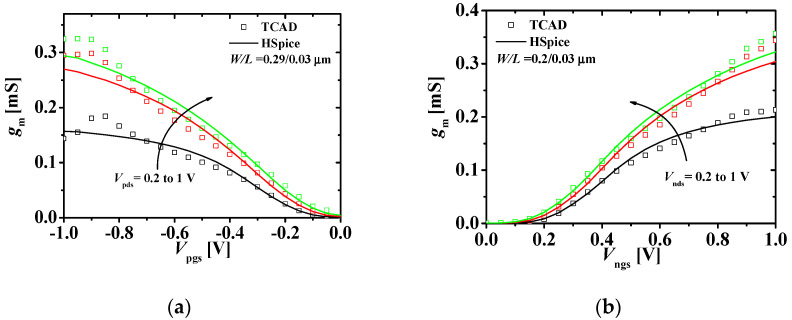
Transconductance-voltage characteristics at different values of *V*_ds_s: (a) the p-type JLFET; and (b) the n-type JLFET (the squares and lines denote the TCAD and HSPICE simulation results, respectively).

**Figure 7 micromachines-11-00887-f007:**
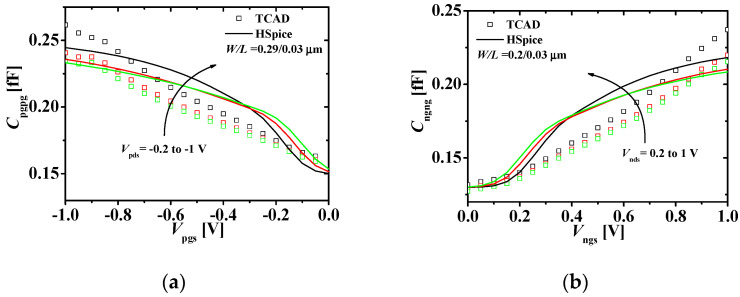
(**a**) Gate capacitance (*C*_pgpg_) of the bottom p-type JLFET at different values of *V*_pds_s; (**b**) gate capacitance (*C*_ngng_) of the top n-type JLFET at different values of *V*_nds_ s and *V*_pgs_ s = 0 V (symbols and lines denote the TCAD and HSPICE simulation results, respectively).

**Figure 8 micromachines-11-00887-f008:**
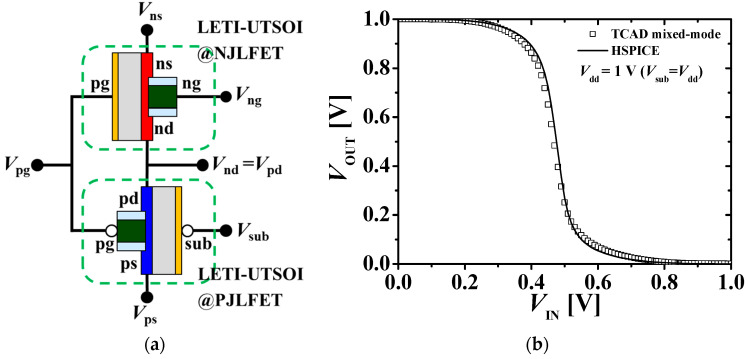
(**a**) Equivalent circuit of M3DINV-JLFET; (**b**) VTC of M3DINV-JLFET; *V*_SS_ = 0 V and *V*_DD_ = 1 V (the symbols and lines denote the TCAD mixed-mode and HSPICE simulation results, respectively; *V*_IN_ = *V*_pg_ = *V*_ng_, *V*_OUT_ = *V*_pd_ = *V*_nd_, and *V*_sub_ = *V*_DD_).

**Figure 9 micromachines-11-00887-f009:**
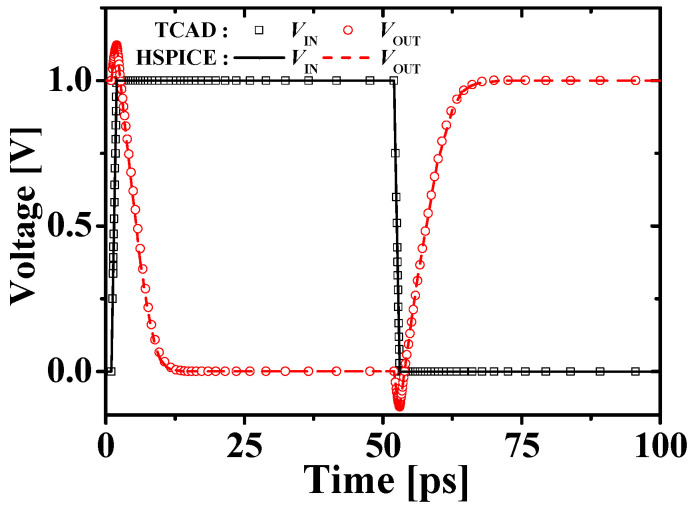
Transient response of the M3DINV-JLFET (symbols and lines denote the TCAD mixed-mode and HSPICE simulation results, respectively; load capacitance *C_L_* = 1 fF).

**Table 1 micromachines-11-00887-t001:** Descriptions and dimensions of the models/parameters used in the TCAD simulation.

Models/Parameters	Description	Value/Unit
CVT	Lombardi model and complete mobility model including the doping density N, temperature T, and transverse electric field E_//_	–
SRH	Shockley-Read-Hall recombination model	–
BGN	Band gap narrowing model	–
AUGER	Auger recombination model	–
FERMI	Fermi-Dirac carrier statistics	–
NEWTON	Newton method which solves a linearized version of the entire nonlinear algebraic system	–
GUMMEL	GUMMEL method, which solves a sequence of relatively small linear subproblems	–
*Φ_N_*	Gate work function of N-type JLFET	5.06 eV
*Φ_P_*	Gate work function of P-type JLFET	4.41 eV

**Table 2 micromachines-11-00887-t002:** Summary of the extracted parameters of the LETI-UTSOI model for the bottom p-type and top n-type JLFET.

Parameter	Unit	Description	Value	
			P-Type	N-Type
DLQ	m	Effective channel length offset for C−V (capacitance −voltage)	2 × 10^−8^	1 × 10^−8^
VBFO	V	Geometry-independent flat-band voltage	0.29	−0.28
CICO	–	Geometry-independent part of substrate bias dependence factor of interface coupling	0.65	3
PSCEL	–	Length dependence of short channel effect above threshold	0.5	0.1
CFL	V^−1^	Length dependence of DIBL (Drain-Induced Barrier Lowering) parameter	2.7	2
UO	m^2^/V/s	Zero-field mobility	6.5 × 10^−3^	1.75 × 10^−2^
MUEO	m/V	Mobility reduction coefficient	1	1
THEMUO	–	Mobility reduction exponent	1.22	1.8
RSGO	–	Gate-bias dependence of RS (Resistance)	2	1
THESATO	V^−1^	Geometry-independent Velocity saturation parameter	1.8	2.7
THESATBO	V^−1^	Substrate bias dependence of velocity saturation	0.1	0.28
FETAO	–	Effective field parameter	0	−3
AXO	–	Geometry-independent of linear/saturation transition factor	1.6	1.6
ALPL1	–	Length dependence of CLM pre-factor ALP	0.0005	0.00001
VPO	V	CLM logarithm dependence factor	0.04	0.04
CFRW	F	Outer fringe capacitance	2 × 10^−16^	2 × 10^−16^

**Table 3 micromachines-11-00887-t003:** Fanout-3 (FO3) ring oscillator performance using M3DINV models (MOSFET and JLFET).

Stages	Power [*μW*]		Frequency [GHz]		Delay Per Stage [*ps*]	
	MOSFET	JLFET	MOSFET	JLFET	MOSFET	JLFET
3	281	275 (−2.13%)	18.7	18.18 (−2.78%)	9.04	9.165 (1.38%)
19	279	277 (−0.71%)	2.88	2.86 (−0.69%)	9.16	9.18 (0.2%)
101	281	283 (0.71%)	0.52	0.52 (0%)	9.28	9.35 (0.7%)

**Table 4 micromachines-11-00887-t004:** Performance comparison of the M3DIC-MOSFETs and M3DIC-JLFETs.

Performances	M3DIC-MOSFET [27]						M3DIC-JLFET					
	INV	NAND	NOR	MUX	D-FF	SRAM	INV	NAND	NOR	MUX	D-FF	SRAM
*Average static power* [*nW*]	4.89	9.84	6.23	11.8	15.4	4.3	10.6 (116.7%)	78 (692%)	54.5 (774%)	104.7 (787%)	123.2 (700%)	20.3 (372%)
*Average dynamic power* [*μW*]	9.85	21.1	16.6	37.7	41.9	20	16.5 (67.5%)	29.3 (38.8%)	20.7 (24.6%)	46.6 (23.6%)	47.8 (14%)	28.2 (41%)
*Average delay* [*ps*]	4.17	4.61	5.65	4.41	10.25	8.1	4.65 (11.5%)	6.1 (32.3%)	7.31 (29.3%)	4.72 (7%)	11.9 (16%)	8.85 (9.25%)

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
