# Peer review of "Circuit Simulation Considering Electrical Coupling in Monolithic 3D Logics with Junctionless FETs"

_micromachines, 2020, doi:10.3390/mi11100887_

Round 1

Reviewer 1 Report

I think some of the figures are not efficient to show the validity and efficiency of the model. Please pay attention to the fact that there are several well-established models, e.g. EPFL JL Model, already available and indeed the authors should prove the effectiveness of their concept in terms of physics-based and compact modeling. It should be noted that UTSOI has not developed for junctionless devices.

The highest doping concentration discussed in this work seems 1e19 cm-3, Could you prove the validity of your approach for higher doping concentration? In junctionless FETs, if the silicon layer is extremely doped (especially in the center of the channel where the peak of doping concentration is located) and/or too thick, it may be unfeasible to fully deplete the silicon channel from carriers and thus to switch off the device. Could you please prove the validity of your results for higher ND?

The analysis is based on a classical transport formulation and, thus, neglects quantum-mechanical effects due to carrier confinement. Nevertheless, to account for the physical mechanisms properly, the Philips unified mobility model, Lombardi model and high field saturation model are required to capture the bulk and surface transport of carriers. Shockley-Read-Hall (SRH), Auger and BTBT models must be included for carrier recombination and tunneling in TCAD simulation results. It seems to me none of those physical phenomena are taken into account in TCAD simulations. Please clarify it. The authors might provide more details of the models used in their TCAD simulations.

In addition, reviewer thinks that the bibliography should be extended and references should be made. Authors may cite other relevant papers and books to provide a more comprehensive background of junctionless for interested readers:
1) Jazaeri, F., & Sallese, J. (2018). Modeling Nanowire and Double-Gate Junctionless Field-Effect Transistors. Cambridge: Cambridge University Press.
2) EPFL Junctionless model

3) Generalized charge-based model of double-gate junctionless FETs, including inversion

4)...

Author Response

I think some of the figures are not efficient to show the validity and efficiency of the model. Please pay attention to the fact that there are several well-established models, e.g. EPFL JL Model, already available and indeed the authors should prove the effectiveness of their concept in terms of physics-based and compact modeling. It should be noted that UTSOI has not developed for junctionless devices.

-> Thanks for your comments. We checked about the EPFL-JL model. In this paper, the asymmetric double gate model is suitable for the M3DI circuit simulation model. However, the EPFL-JL model is identified as a symmetric double gate model. It is not suitable as a SPICE model that can consider the electric coupling of the proposed M3DI structure. Currently, the EPFL-JL model does not have an open source code such as Verilog-A model. Therefore, we referred to the fact that the circuit simulation using BSIM-CMG model for GAA JLFET were performed in Ref. A-1. Using the MOSFET-based LETI-UTSOI model, we tried to verify that it is possible to simulate a circuit considering the electric coupling of M3DIC-JLFET.

[A-1] Chang, S.-W.; Sung, P.-J.; Chu, T.-Y.; et al. First Demonstration of CMOS Inverter and 6T-SRAM Based on GAA CFETs Structure for 3D-IC Applications. . In Proceedings of 2019 IEEE International Electron Devices Meeting (IEDM), San Francisco, CA, USA, 7-11 December 2019, pp. 11.7.1-11.7.4, doi: 10.1109/IEDM19573.2019.8993525.

The highest doping concentration discussed in this work seems 1e19 cm-3, Could you prove the validity of your approach for higher doping concentration? In junctionless FETs, if the silicon layer is extremely doped (especially in the center of the channel where the peak of doping concentration is located) and/or too thick, it may be unfeasible to fully deplete the silicon channel from carriers and thus to switch off the device. Could you please prove the validity of your results for higher ND?

-> Thanks for your comments. The significance of this paper is the possibility of circuit simulation considering the electric coupling of M3DIC-JLFET. LETI-UTSOI is a low-doped ultrathin body and buried oxide fully depleted silicon-on-isolator transistor model with strong forward back bias. The JLFET was set to a typical thin (Tsi = 6 nm) silicon and a doping concentration of 1e19 cm-3. Coupling for JLFETs with higher ND will be studied in next.

The analysis is based on a classical transport formulation and, thus, neglects quantum-mechanical effects due to carrier confinement. Nevertheless, to account for the physical mechanisms properly, the Philips unified mobility model, Lombardi model and high field saturation model are required to capture the bulk and surface transport of carriers. Shockley-Read-Hall (SRH), Auger and BTBT models must be included for carrier recombination and tunneling in TCAD simulation results. It seems to me none of those physical phenomena are taken into account in TCAD simulations. Please clarify it. The authors might provide more details of the models used in their TCAD simulations.

-> Thanks for your comments. In this paper, the Lombardi CVT model was used as the mobility model and the SRH (Shockley-Read-Hall) and Auger model were used as the recombination model for the TCAD simulation. In addition, BGN (Band gap narrowing) model and FERMI model were used. According to the ATLAS manual in Silvaco, CVT models do not allow combinations with other mobility models. However, the combination with the high field mobility model (FLDMOB) was not allowed, and the Philips unified mobility model (KLASSEN) was allowed with the CVT model. Figure A-1 shows the difference between the current model and the added KLASSEN model. there is no significant difference with the error within 10%. Since the error is small, the simulation using the current model is enough to investigate the electrical coupling between the top N-type and bottom P-type JLFETs, but for an accurate simulation, all of models suggested by you will be used in additional studies.

Figure A-1. Comparison of current-voltage characteristics by adding mobility model

In addition, reviewer thinks that the bibliography should be extended and references should be made. Authors may cite other relevant papers and books to provide a more comprehensive background of junctionless for interested readers:

1) Jazaeri, F., & Sallese, J. (2018). Modeling Nanowire and Double-Gate Junctionless Field-Effect Transistors. Cambridge: Cambridge University Press.

2) EPFL Junctionless model

3) Generalized charge-based model of double-gate junctionless FETs, including inversion

4)...

-> According to your suggestions, we added several references.

Reviewer 2 Report

The authors proposed “Circuit Simulation Considering Electrical Coupling in Monolithic 3D Logics with Junctionless FETs” by TCAD and HSPICE simulation. The authors merely discuss the simulation results rather than the detailed mechanism on IdVg, IdVd, and inverter of JLFETs. Based on the HSPICE simulation, authors could clarify whether the simulated JLFETs results would be influenced by parameters such as dopants fluctuation, activation temperature, and channel doping distribution. Suggestion on revisions were listed as follows:

  1. Fig. 5 (a) shows the HSPICE results with 10% error as compared with TCAD results. Explanation on top n-type JLFET in Fig. 5 (a) with more Inds dispersion than Fig. 4(a) is necessary.
  2. Fig. 8(b) shows voltage transfer characteristics in p-type and n-type JLFETs, and reveals symmetricity in the threshold voltage. However, the dopants fluctuation might degrade on the Vth variation and it should be considered in simulation for inverter.
  3. Fig.1 shows the schematic of a JLFET-constructed M3D inverter in which the NMOS (top tier) and PMOS (bottom tier) were connected by MIV interconnect. Therefore, the influences by the resistance of MIV should be addressed.
  4. Could the Author add the 6T SRAM bit cell information in the table 4?
  5. Why is the average delay of NAND in M3DIC-JLFET with an increase by 21.4% is quite different from the rest of the data (static power, Dynamic power) by only 2.45% and 2.15% ?
  6. Why is the static power of NOR in M3DIC-JLFET with an increase by 29.4% is quite different from he rest of the data (average delay, Dynamic power) by only 1.34% and 3.62% ?

Author Response

The authors proposed “Circuit Simulation Considering Electrical Coupling in Monolithic 3D Logics with Junctionless FETs” by TCAD and HSPICE simulation. The authors merely discuss the simulation results rather than the detailed mechanism on IdVg, IdVd, and inverter of JLFETs. Based on the HSPICE simulation, authors could clarify whether the simulated JLFETs results would be influenced by parameters such as dopants fluctuation, activation temperature, and channel doping distribution. Suggestion on revisions were listed as follows:

Fig. 5 (a) shows the HSPICE results with 10% error as compared with TCAD results. Explanation on top n-type JLFET in Fig. 5 (a) with more Inds dispersion than Fig. 4(a) is necessary.

-> Thanks for your comments. The structure of M3DIV-JLFET in this paper has an ILD of 10 nm, so the distance between the gate of the bottom p-type JLFET and the silicon of the top n-type JLFET is very short. Based on the top n-type JLFET, you can think of it as an independent multi-gate JLFET structure. Fig. 5 (a) checks whether the current from the front-gate (gate of n-type JLFET) reflects the Vth change well due to the application of the back-gate (gate of p-type JLFET) voltage. Fig. 4 (a) is the current-voltage characteristics of the top n-type JLFET in a state where there is no back-gate effect (VBG = 0 V). We modified content for the Fig. 5 in lines 109-111.

Fig. 8(b) shows voltage transfer characteristics in p-type and n-type JLFETs, and reveals symmetricity in the threshold voltage. However, the dopants fluctuation might degrade on the Vth variation and it should be considered in simulation for inverter.

-> Thanks for your comments. LETI-UTSOI is the available model which can describe the behavior of low-doped ultrathin body and buried oxide fully depleted silicon-on-insulator transistors in all bias configurations, including strong forward back bias. Since it is a low-doped model, there is no parameter to set the doping of the channel. The LETI-UTSOI model cannot take dopants fluctuation into account in circuit simulation.

Fig.1 shows the schematic of a JLFET-constructed M3D inverter in which the NMOS (top tier) and PMOS (bottom tier) were connected by MIV interconnect. Therefore, the influences by the resistance of MIV should be addressed.

-> Thanks for your comments. The final structure includes the effect of R/C by MIV/ML as shown in Figure 1. In [A-2], the effects of MIV/ML were studied. In this paper, since the main concern is to propose a model for circuit simulation considering the electric coupling of M3DINV-JLFET, we plan to research on the entire M3DIC-JLFET including the effects of MIV/ML in next.

[A-2] Ahn, T.J.; Perumal, R.; Lim, S.K.; Yu, Y.S. Parameter Extraction and Power/Performance Analysis of Monolithic 3-D Inverter (M3INV). IEEE Trans. Electron Devices. 2019, 66, 1006–1011, doi: 10.1109/TED.2018.2885817.

Could the Author add the 6T SRAM bit cell information in the table 4?

-> According to your suggestions, we added 6T SRAM bit cell information in Table 4.

Why is the average delay of NAND in M3DIC-JLFET with an increase by 21.4% is quite different from the rest of the data (static power, Dynamic power) by only 2.45% and 2.15% ?

-> Thanks for your comments. There was a mistake in measuring static power, dynamic power, and average delay in Table 4. The inputs of the inverter are low (0) or high (1), but the inputs of NAND and NOR are (0,0), (0,1), (1,0), and (1,1). Because we did not consider all cases of inputs, the static power, dynamic power, and delays are modified as the average static power, average dynamic power, and average delay for each input case.

M3DIC-JLFETs have more leakage current changes due to electrical coupling than M3DIC-MOSFETs. Electrical coupling by the gate of bottom transistor increases the leakage current of the top transistor, resulting in an increase in static power. Due to electric coupling, the load cap of M3DIC-JLFETs is larger than that of M3DIC-MOSFETs, so the dynamic power and delay of M3DIC-JLFETs are larger. We modified content for Table 4 in lines 165-168 and 170-172.

Why is the static power of NOR in M3DIC-JLFET with an increase by 29.4% is quite different from the rest of the data (average delay, Dynamic power) by only 1.34% and 3.62% ?

-> Thanks for your comments. There was a mistake in measuring static power, dynamic power, and average delay in Table 4. The inputs of the inverter are low (0) or high (1), but the inputs of NAND and NOR are (0,0), (0,1), (1,0), and (1,1). Because we did not consider all cases of inputs, the static power, dynamic power, and delays are modified as the average static power, average dynamic power, and average delay for each input case. M3DIC-JLFETs have more leakage current changes due to electrical coupling than M3DIC-MOSFETs. Electrical coupling by the gate of bottom transistor increases the leakage current of the top transistor, resulting in an increase in static power. Due to electric coupling, the load cap of M3DIC-JLFETs is larger than that of M3DIC-MOSFETs, so the dynamic power and delay of M3DIC-JLFETs are larger. We modified content for Table 4 in lines 165-168 and 170-172.

Round 2

Reviewer 1 Report

All my comments have been addressed in the revised manuscript and I recommend its publication.

Author Response

Thanks for your recommendation.

Yun Seop Yu

Reviewer 2 Report

  1. Information in table 3 and table 4 is not coordinate. Why is the increase in power consumption of 101 stages ring oscillator (Table 3) lower than that for single inverter (Table 4)? Moreover, why is the average inverter delay (Table 4) is longer than that for 101 stages ring oscillator? Suggest authors to propose an explanation.

Author Response

Information in table 3 and table 4 is not coordinate.

-> They are consistent. Please check the following answers.

Why is the increase in power consumption of 101 stages ring oscillator (Table 3) lower than that for single inverter (Table 4)?

- Thanks for your comments. The power consumption of the single-stage inverter consisting of M3DIC-MOSFETs and JLFETs in Table 4 is 9.854 and 16.51 uW, respectively, which is lower than those of 101 stages ring oscillator consisting of the M3DIC-MOSFET and JLFET in Table 3 (281 and 283 uW, respectively).

Moreover, why is the average inverter delay (Table 4) is longer than that for 101 stages ring oscillator?

-> Thanks for your comments. The average delays of the single-stage inverter consisting of M3DIC-MOSFETs and JLFETs in Table 4 are 4.17 and 4.65 ps, respectively, which is lower than those of 101 stages ring oscillator consisting of M3DIC-MOSFETs and JLFETs in Table 3 (9.28 and 9.35 ps, respectively).